



# Simulating the radiative forcing of oceanic dimethylsulfide (DMS) in Asia based on Machine learning estimates

Junri Zhao[1,2], Weichun Ma[1,3], Kelsey R Bilsback[4], Jeffrey R Pierce[4], Shengqian Zhou[1,2], Ying Chen[1,2], Guipeng Yang[5], Yan Zhang[1,2,3,6*]

[1]Shanghai Key Laboratory of Atmospheric Particle Pollution and Prevention (LAP3), Department of Environmental Science and Engineering, Fudan University, Shanghai 200438, China
[2]Shanghai Institute of Eco-Chongming (SIEC), Shanghai 200062, China
[3]Institute of Digitalized Sustainable Transformation,Big Data Institute, Fudan University, Shanghai 200433, China
[4]Department of Atmospheric Science, Colorado State University, Fort Collins, CO, United States of America
[5]Key Laboratory of Marine Chemistry Theory and Technology, Ministry of Education, College of Chemistry and Chemical Engineering, Ocean University of China, Qingdao 266100, China
[6]National Observations and Research Station for Wetland Ecosystems of the Yangtze Estuary,Shanghai, China

*Correspondence to*: Yan Zhang (yan_zhang@fudan.edu.cn)

**Abstract.** DMS emitted from the sea water is a key precursor to new particle formation, acting as a regulator in Earth's warming climate system. However, DMS's effects are not well understood in various ocean regions. In this study, we estimated DMS emissions based on a machine learning method and used the GEOS-Chem global 3D chemical transport model coupled with the TwO Moment Aerosol Sectional (TOMAS) microphysics scheme to simulate the atmospheric chemistry and radiative effects of DMS. The contributions of DMS to atmospheric $SO_4^{2-}$ aerosol and cloud condensation nuclei (CCN) concentrations

along with their radiative effect over the Asian region were evaluated for the first time. Firstly, we constructed novel monthly-resolved DMS emissions ($0.5° \times 0.5°$) for the year 2017 using a machine learning model. 4351 seawater DMS measurements (including the recent ones over the Chinese Sea) and 12 relevant environment parameters were selected for training. We found the model could predict the observed DMS concentrations with a correlation coefficient of 0.75 and fill the values in regions lack of observations. Across Asian Seas, the highest seasonal mean DMS concentration occurred in Mar-Apr-May (MAM),

and we estimate annual DMS emission flux of 1.25 Tg (S), which accounts for 15.4% of anthropogenic sulfur emissions over the entire simulation domain (covers most of Asia) in 2017. The model estimates of DMS and methane sulfonic acid (MSA), using updated DMS emissions, were evaluated by comparing with cruise survey experiments and long-term online measurement site data. The improvement in model performance can be observed compared with the global-database DMS emissions. The relative contributions of DMS to $SO_4^{2-}$ and CCN were higher in remote oceanic areas, which reached up to 88%

and 42% of all sources. Correspondingly, the sulfate direct radiative forcing (DRF) and indirect radiative forcing (IRF) contributed by DMS ranged from -200 to -20 mW m$^{-2}$ and -900 to -100 mW m$^{-2}$, respectively, with levels varying by season. The strong negative IRF is mainly over remote ocean regions ( -900 to -600 mW m$^{-2}$). Generally, the magnitude of IRF derived





by DMS was twice as large as its DRF. This work provides insights into the source strength of DMS and its impact on climate, addressing knowledge gaps related to factors controlling aerosols in the marine boundary layer and their climate impacts.

## 1 Introduction

Ocean-emitted DMS is a precursor of non-sea-salt $SO_4^{2-}$ and controls the composition, size distribution, and number concentrations of aerosols over the remote oceanic areas. $SO_4^{2-}$ directly influences the climate system by reflecting solar radiation back into the space, and indirectly, acting as CCN and altering the albedo of clouds and changing cloud radiative properties (Andreae and Rosenfeld, 2008). The "CLAW" hypothesis proposed by Charlson et al. (1987) assumed that negative feedback interactions between ocean plankton and climate system, where the Earth system acted to buffer itself from warming, was linked through DMS production. Thereafter, several studies found that significant impacts of DMS induced aerosols on CCN and cloud albedos in remote oceans (Park et al., 2017;Quinn et al., 2017;Kulmala et al., 2014;Vallina and Simó, 2007), which lend credence to CLAW hypothesis. Nevertheless, due to low sensitivity of each step of the interactions to changes in force factors in the CLAW climate feedback loop (e.g., low sensitivity of DMS production to changes in incident solar radiation), Quinn and Bates (2011) disproved the hypothesis. Whether the CLAW climate feedback is positive or negative is still uncertain, and further research is need to quantify the climate effect of DMS.

Building an accurate emission inventory is key to simulating the climate effects of DMS. As many previous studies have shown (Chen et al., 2018;Hodshire et al., 2019;Rap et al., 2013;Yang et al., 2017;Zhao et al., 2021), the marine DMS emissions used in numerical models mainly were estimated using an interpolation scheme (Kettle et al., 1999;Lana et al., 2011), which mostly estimated DMS climatology by interpolating observed DMS data in limited sites to the global ocean. At first, observations from Global Surface Seawater DMS Database were binned and grouped into 57 ecological geographic ocean provinces, with weighted interpolations from near provinces used to fill the values where without observations. Wang et al., (2020) has pointed out that there were uncertainties in using spatial and temporal averaged data to fill regions without observations. However, artificial neural networks can potentially be trained and used to fill measurement gaps (Wang et al., 2020). Galí et al. (2018) created a remote sensing algorithm to estimate DMS concentrations which is based on the relationship between a precursor of DMS and plankton light exposure. Their results (Galí et al., 2018) indicated that the remote sensing algorithms have better ability to reproduce the climatological features of DMS seasonality than interpolated DMS climatologies, which also outweigh the disadvantage of the interpolation scheme used in previous study (Lana et al., 2011). In a recent study (Bell et al., 2021), long-term in-situ DMS measurements conducted in North Atlantic Ocean from 2015 to 2018 were compared with the interpolated DMS climatologies (Lana et al., 2011) and predicted DMS concentrations from the remote sensing algorithm (Galí et al., 2018) and the neural network approach (Wang et al., 2020), which revealed that that both the remote sensing algorithm and the neural network model reproduce the sea water DMS trends better than the interpolated climatologies. However, DMS predictions from two model(Galí et al., 2018;Wang et al., 2020) tend to be underpredicted, likely because the primary biological processes of DMS production was not accounted for(Bell et al., 2021).





There are several modelling studies which have quantified the aerosol direct and indirect radiative forcing of DMS on a global scale. The global annual mean DMS aerosol indirect radiative forcing estimates have ranged from -6.55 to -0.23 W/m$^2$ (Mahajan et al., 2015;Thomas et al., 2010;Rap et al., 2013;Yang et al., 2017;Jin et al., 2018). There have been few studies that have reported the radiative effect of DMS on a regional scale. Choi et al. (2020) adopted an empirical algorithm to estimate DMS concentrations and calculated direct radiative effect of DMS aerosol, that is -1.3 W/m$^2$ for the year 2014-2016 over East

Asian seas, which was even higher than those global results (Yang et al., 2017;Rap et al., 2013). And there were no evaluations of the DMS predictions in the seawater and atmosphere in their studies, which leads to an unknown reliability of the results. The annual-mean direct radiative forcing due to DMS produced aerosol were -0.2 to -0.1 W/m$^2$ over East Asia reported by Li et al. (2019) who used a DMS climatology(Lana et al., 2011) with $1° \times 1°$ horizontal resolution for radiative forcing calculation. As mentioned before, some uncertainties in the DMS climatology estimated by an interpolation scheme and coarse grid ($1° \times$

$1°$) may not be appropriate for regional simulations. In the previous studies (Li et al., 2020b;Li et al., 2020a), we used long-term DMS measurements in 2011, 2013, 2015, 2016, and 2017 from a series of shipboard field experiments and performed interpolation to map DMS concentrations in Chinese seas. The newest DMS measurements were used to explore the impact of DMS on air quality over coastal areas of China, but the radiative effect of DMS was not reported.

To our knowledge, this is the first systematic study of the Asia region that quantifies the impacts of DMS on sulfate,

particle number concentration, and radiative forcing by using state-of-the-art aerosol microphysics model coupled on global 3D chemical transport model. In this study, we first trained 12 environmental parameters and newly updated DMS measurements by utilizing eXtreme Gradient Boosting (XGBoost) machine learning algorithms (Chen and Guestrin, 2016) used to construct a new regional DMS emissions for the year 2017 with a plausible underlying basis in ocean environmental parameters. Then, the model performances of DMS and MSA were evaluated by comparing the model simulations with

shipboard field measurements and long-term online measurement site data. Finally, the annual-average and seasonal impacts of DMS on sulfate/CCN concentrations and direct/indirect radiative forcing were quantified.

## 2 Methods and data

### 2.1 GEOS-Chem-TOMAS

In this study, the GEOS-Chem version 12.9.3 (https://doi.org/10.5281/zenodo.3974569, last access: 25 March 2021) cou pled with the online TOMAS aerosol microphysics model (Adams and Seinfeld, 2002) was adopted to calculate atmospheric aerosol size, number, and mass concentrations from marine DMS emissions. TOMAS was used to simulate aerosol microphy sics processes (i.e., nucleation, coagulation, condensation, cloud processing). The advantage of TOMAS is the full size resol ution for all chemical species and the conservation of aerosol number, which allows modeler to construct aerosol and CCN n

umber budgets that will balance. GEOS-Chem-TOMAS (GC-TOMAS) has been used in a range of previous studies (Kodros and Pierce, 2017;Pierce and Adams, 2006;Kodros et al., 2016;D'Andrea et al., 2013;Westervelt et al., 2013;Lee et al., 2009;T



rivitayanurak et al., 2008;Pierce et al., 2007;Adams and Seinfeld, 2002). The model contains detailed hydrocarbon – nitrogen oxide ($NO_x$) – ozone($O_3$) - volatile organic compounds(VOC) – bromine oxides($BrO_x$) tropospheric chemistry (Bey et al., 2001) and aerosol species (including sulfate, nitrate, ammonium, black carbon, organic carbon, mineral dust, and sea salt) (Duncan Fairlie et al., 2007;Pye et al., 2009;Alexander et al., 2005;Park et al., 2004) that are fully coupled to gas-phase chemistry, with the ISORROPIA II algorithm to calculate the thermodynamic equilibrium between aerosols and their gas phase precursors (Fountoukis and Nenes, 2007). The model includes detailed wet and dry deposition scheme for aerosols and gas species which have been described in previous studies (Wesely, 2007;Liu et al., 2001;Wang et al., 1998;Amos et al., 2012). This version of GC-TOMAS tracks the total aerosol particle number and the mass of each aerosol species (sulfate, mineral dust, sea salt, hydrophilic and hydrophobic organic carbon, externally and internally mixed elemental carbon, and aerosol water) across 15 logarithmically size bins ranging from 3 nm to 10 μm (Lee and Adams, 2012;Lee et al., 2013). Since the ammonium nitrate size distribution is not explicitly tracked with GC-TOMAS, so we assume that it follows the aerosol water distribution(Bilsback et al., 2020a;Bilsback et al., 2020b).

The simulation domain covers most of Asia (11°S to 55°N, 60–150°E), discretized with a horizontal grid resolution of 0.5° × 0.625° and 47 vertical layers, and uses Modern-Era Retrospective Analysis for Research and Applications Version (MERRA-2) assimilated meteorological field for meteorological inputs (Gelaro et al., 2017). To assess radiative impacts of DMS emissions at a regional scale, we performed three different annual simulations for the year 2017 (Table 1). The XG simulation represents DMS emissions were calculated from our updated DMS emissions estimates (see Section 2.3) and the other (LANA) refer to DMS emissions from Lana DMS climatology (Lana et al., 2011), which is default setting in current version of Geos-Chem model. The ND simulation represents DMS emissions tuned off. Each simulation was conducted with 1 month spin-up period (December 2016). The boundary conditions for the simulation domain were obtained from global simulations at 2° × 2.5° with 47 vertical layers.

For anthropogenic emissions in Asia, we used the recently updated Global anthropogenic emission inventories (0.5°x0.5°) or the year 2017 from the open-source Community Emissions Data System (CEDS) (McDuffie et al., 2020), which applied scale factors from Zheng et al., (2018) to update China's emissions for the year 2017. Since there is a significant reduction (62%) in $SO_2$ emissions in China from 2010 to 2017 (Zheng et al., 2018), updated emissions for China are crucial for quantifying contributions of biogenic sulfur source over Asia. Biomass burning emissions in the GC-TOMAS are obtained from Global Fire Emissions Database Version 4 (van der Werf et al., 2017). Dust, biogenic VOCs, sea salt, soil $NO_x$, and lighting $NO_x$ emissions are calculated online based on MERRA-2 meteorological field. The Dust Entrainment and Deposition (DEAD) scheme from Zender et al., (2003) was implemented GEOS-Chem to simulate dust mobilization. The Model of Emissions of Gases and Aerosols from Nature from Guenther et al., (2012) was used to generate biogenic VOCs emissions. Soil and lighting $NO_x$ emissions are calculated by parameterization scheme described in Hudman et al., (2012) and Price and Rind (1992), respectively.

The sea-air flux of DMS is estimated using the following the empirical formula as described in (Lana et al., 2011):

$$F = C_w \times k_w \times (1-\gamma) \tag{1}$$



Where, $C_w$ is the seawater DMS concentrations and $k_w$ is the water side gas transfer velocity and $\gamma$ is the atmospheric gradient fraction. In this study, we selected the Nightingale et al. (2000) parameterization (hereafter N00) for $k_w$ to represent the DMS emissions over the global ocean.

## 2.2 Radiative forcing calculation scheme

To calculate the top-of-atmosphere (TOA) all sky DRF and cloud-albedo IRF, we used Rapid Radiative Transfer Model for Global Climate Models (RRTMG) (Iacono et al., 2008) with monthly averaged aerosol number and mass concentrations from GC-TOMAS output and meteorological variables from MERRA2. For the DRE, we calculated aerosol optical depth (AOD) single scattering albedo, and the asymmetry parameter based on Mie theory (Bohren and Huffman, 1983) and refractive indices from the Global Aerosol Database (Koepke et al., 1997). In all cases, the DRE was calculated for core-shell optical assumption, where, for each aerosol size bin, black carbon was represented as a spherical core within a homogenous shell of all other hydrophilic species. For the cloud-albedo IRF, we calculate cloud droplet number concentration (CDNC) using the activation parameterization from Abdul-Razzak and Ghan (2002). Cloud-liquid water content is prescribed from MERRA-2 and held fixed, and hence we only calculated the cloud-albedo (Twomey) indirect effect. The changes in effective cloud drop radii were estimated following the cloud-droplet-radius perturbation method used in previous studies (Rap et al., 2013;Kodros et al., 2016;Scott et al., 2014). Then, RRTMG is used to calculate the changes of TOA radiative flux from the changes effect cloud drop radii, and we limit this calculation to liquid clouds, which is a limitation in this method. More detailed information about implementation of RRTMG in GC-TOMAS can be found in Kodros et al., (2016).

## 2.3 Machine learning estimates of sea-surface DMS concentration for calculating DMS emission flux

To better capture the nonlinear relationship between DMS and its influencing parameters, we trained XGBoost model (machine learning algorithm under the Gradient Boosting framework) with the entire dataset to predict sea surface DMS concentrations where without the observations. Figure S1 shows the spatial distribution of DMS measurements. The red points (1022 valid measurements) represent local DMS observations dataset (2011, 2013, 2015, 2016, and 2017) in Chinese seas from China Ocean University. Details can be found in our previous studies (Yang et al., 2015a;Yang et al., 2014;Yang et al., 2015b;Xu et al., 2021;Zhai et al., 2020;Wu et al., 2020;Jian et al., 2019;Yu et al., 2019;Mao et al., 2021). The blue points (3329 valid measurements) represent the observations from Global Surface Seawater DMS Database (http://saga.pmel.noaa.gov/dms/; last access: 1 May 2021). In total, 12 environmental parameters (Table S1) which strongly affect the growth of phytoplankton and the production of DMS (Wang et al., 2015) were included as predictors in machine learning estimates. Satellite remotely sensed Chlorophyll (Chl), Photosynthetically available Radiation (PAR), Particulate inorganic/organic Carbon (PIC/POC), and Diffuse attenuation coefficient at 490m (kd490) were from MODIS-Aqua products (daily,8-day, and monthly Level 3-binned 4km resolution data). Nutrient data (Silicate, Phosphate, and Nitrate), Sea surface





temperature (SST), and Dissolved Oxygen (DO) were obtained from World Ocean Atlas 2018 (monthly 0.25° and 1° climatology data). Monthly mixed layer depth (MLD) climatology (0.5° × 0.5°) was obtained from Monthly Isopycnal &

Mixed-layer Ocean Climatology (MIMOC). Before the implementation of the algorithm to Asia's oceans, we performed a model validation. Firstly, the environmental parameters were matched with DMS measurements according to sampling geographical coordinates and date. Take remotely sensed Chl data, for example, if the daily binned data failed to match the DMS observed data, we used the 8-day binned data to take the place of daily binned data. After the data matching, we then conducted filtering and quality control which followed methods from Wang et al., (2020), the number of data points in the

simulation domain was reduced from 4351 to 3748 observation-based datasets for in-situ DMS and matched with environmental parameters. Table S1 has a description of the environmental parameters, sources, and their filtering thresholds. To verify the prediction performance of XGBoost model, we divided the datasets into two parts: validation datasets and training datasets. Considering that most of the northern part of the simulation domain was land area, we selected the data from 2° latitude bands between 11°S and 30°N as validation datasets (809 points), while the rest of the data was all used as training

data (2939 points).

Figure 1 displays the validation results for XGBoost model, which reproduced DMS concentrations with high correlation coefficients (R) of 0.75 and low root-mean-square error (RMSE) of 1.97 µmol m$^{-3}$. The validation statistics are comparable to other studies (R=0.73-0.81 and RMSE=1.92 -2.00 µmol m$^{-3}$) that used nonlinear/multilinear models to predict sea-surface DMS concentrations over the global ocean (Galí et al., 2018;Wang et al., 2020). The advantage of utilizing machine learning

method is that the ability of capturing nonlinear relationships between DMS and its affecting parameters to estimate DMS concentrations with a plausible underlying basis in spatial-temporal variability. A shortcoming of the traditional geographical interpolation method is that relatively sparse data is typically interpolated to the entire ocean, which has been highlighted by previous studies (Galí et al., 2015;Galí et al., 2018;Wang et al., 2020). In this study, the advantage of the machine learning method is also demonstrated by comparing two different model simulations (see Section 3.2). In the implementation phase of

the machining learning algorithm to regional ocean, to ensure the model was representative of present-day (2017) atmospheric conditions, in addition to climatology environmental parameters, the remotely sensed datasets used to predict DMS concentrations are all from MODIS-Aqua products in 2017. Monthly climatologies were interpolated to the 8-day or monthly periods remotely sensed data; then, we trained XGBoost model to obtain grid values that did not have DMS measurements. Finally, estimated DMS concentration were temporally averaged to a seasonal period and spatially binned to 0.5°×0.5° grid

for Asia region (see Section 3.1).



## 3 Results

### 3.1 Spatial and temporal patterns of the sea-water DMS

Regional DMS maps for sea surface DMS concentrations predicted by XGBoost in four seasons are displayed in Figure
2, which showed distinct seasonal variations. The highest regional mean DMS concentrations were observed in the MAM, that
is 2.52 µmol m$^{-3}$, approximately 1.15, 1.24, and 1.31 times higher than those in Jun-Jul-Aug (JJA), Sep-Oct-Nov (SON), and
Dec-Jan-Feb (DJF) (Table S2), respectively. However, according to the previous studies (Lana et al., 2011;Galí et al.,
2018;Wang et al., 2020), the highest DMS concentrations usually occurred in JJA, mainly attributed to adequate solar
irradiation and warm temperature being favourable for primary production. We assumed that this difference was caused by
different statistical region. Previous results were based on global scale estimates, for comparative purposes, we extracted
corresponding simulation domain (Figure 2) estimates values from global scale estimates results, and they were listed in Table
S2. Across the Asian Seas, all the highest seasonal mean DMS concentrations occurred in MAM, and our estimates agreed
well with the estimates of 2.21-2.33 µmol m$^{-3}$ reported in previous studies (Wang et al., 2020;Lana et al., 2011). As shown in
Figure S2, zonal mean DMS concentrations between 10°S and 30°N latitude areas of simulation domain were higher in MAM
than in JJA, but those between the 30°N and 50°N latitude band were higher in JJA than in MAM. As mentioned in Section
2.3, most of ocean area is concentrated in 10°S and 30°N latitude band of the entire simulation domain (11°S to 55°N, 60–
150°E), which leads to the highest regional mean DMS concentrations being observed in MAM. This is most likely due to the
seasonal variation of solar irradiation, because most of ocean area (11°S to 30°N) in the simulation domain was influenced
more by the solar irradiation in the MAM than in JJA. A similar result can be found in monthly Hovmöller diagrams of DMS
climatologies, depicted by Galí et al. (2018). Throughout the four seasons, there were some high concentrations of DMS
(higher than 4.3 µmol m$^{-3}$) that appeared in different coastal areas, which is probably relevant to high nutrient and chlorophyll
concentrations over the coastal areas. Galí et al. (2015) also found that most of the coastal regions have higher DMSPt
concentrations compared to the global ocean, and DMS in the sea water was generated from the breakdown of DMSPt.

We calculated regional sea-air DMS fluxes using the N00 gas transfer velocity and DMS concentrations predicted by
XGBoost (Figure 3a). We estimated annual DMS emission fluxes of 1.25 Tg (S), which is 15.4% of the anthropogenic sulfur
emissions over the entire simulation domain (covers most of Asia) in 2017. The higher estimated values of DMS fluxes (higher
than 250 tonnes (S)/grid) occurred over some coastal waters, which generally agreed well with the estimated sea surface DMS
concentration distribution. The highest emission fluxes occurred over the Chinese Seas (reach up to 450 tonnes (S)/grid). These
high fluxes can be attributed to local DMS observations dataset in Chinese seas (red point in Figure S1) that were included in
the machine learning estimates. Our previous studies (Li et al., 2020a;Li et al., 2020b) have reported that DMS emissions
fluxes calculated with the local dataset is 3 times higher than default global-database (Lana et al., 2011) over most area of
Chinese Sea. The highest positive changes of DMS emissions fluxes were mainly in the areas of East China Sea (up to 200
tonnes (S)/grid), and some coastal regions (Figure 3b). However, there are more negative changes of DMS emissions fluxes





than positive changes in the sea water, which suggested that sea-air DMS flux estimated in this study generally lower than

those from Lana et al. (2011), and the similar result can be found in Wang et al. (2020).

## 3.2 Model evaluation

### 3.2.1 Model performance of DMS and its oxidation product MSA

Modelled atmospheric DMS concentrations were compared to observations from 2017 Cruise Survey Experiment (CSE) 1-3 (Figure 4). Due to the discontinuities in time and gaps in observations, we averaged the whole period of each CSE

observation for our comparisons. The results in Table S3 demonstrate moderate improvements in the model performance of DMS predictions when using updated DMS emissions relative to default DMS emissions, i.e., the difference between the observations and predictions (observation - prediction) became smaller (from -16.34 to 6.68 pptv for CSE 1, -21.11 to -16.17 pptv for CSE 2, and -121.57 to 117.39 pptv for CSE 3, respectively). CSE 3 had much higher DMS concentrations, because most of the measurements were from the mouth of the Changjiang River, and it is difficult for a coarse model grid ($0.5° \times$

$0.625°$) to represent the high values that occur off coastal areas. MSA is a tracer of DMS, because it is formed exclusively from DMS (Gondwe et al., 2003). We also evaluated the model performances for MSA by comparing the model simulations with long-term online measurement site data (Zhou et al., 2021) from Hua Niao Island (Figure 4). Figure 5 displays time series of daily mean MSA values of predictions (XG and LANA) and observations. The simulated MSA concentrations from XG and LANA are both within the range of observed values, and the trends of the MSA concentrations were relatively well

reproduced, with mean values of 0.014, 0.020, and 0.023 $\mu g\ m^{-3}$ for LANA, XG, and observations. During the period of Jun 21 to Jun 25 and Jun 28 to July 3, LANA simulation results were closer to the observations, and XG simulations underpredicted the measurements in those two periods. Overall, the simulation results of XG in other periods were closer to the observations than those of LANA simulation results.

### 3.2.2 Model performance evaluation for PM$_{2.5}$, AOD, and CCN

The magnitude and distributions of PM$_{2.5}$, AOD, and CCN directly influence DRF and IRF estimates. To evaluate whether GC-TOMAS can reproduce the spatial distribution and temporal trends of these parameters over simulation area, we evaluated model performance by comparing simulation results for XG with ground observations and satellite-retrieved estimates. Since the impacts of DMS to PM$_{2.5}$ and CCN are over the ocean and some coastal areas (see Section 3.3), and the ground observational data is all over land areas, so we only used one of the simulation results for model evaluation.

Boylan and Russell (2006) suggested that when the model performance within the range (mean fractional bias(MFB) $\leq \pm 30\%$ and mean fractional error (MFE) $\leq \pm 50\%$), model predictions can be regarded as sufficiently accurate. Figure S3 presents the distributions of simulated annual mean PM$_{2.5}$ concentrations and observations at 366 city sites from China National Environmental Monitoring Center (CNEMC). The model performed well against PM$_{2.5}$ observations for the year 2017, with MFB of 5.5% and MFE of 23.1%, both within the goal range, and had a Pearson's correlation coefficient (R) of 0.62. Simulated





PM$_{2.5}$ concentrations were slightly underpredicted with a MB of -1.3 µg m$^{-3}$, which is probably ascribed to underpredicted PM$_{2.5}$ in some northern China. Uncertainties in land-based emission inventories tend to cause different model performance in different regions.

Table S4 summarizes the collected in-situ measurements of CCN concentrations in other previous studies and corresponding annual-mean simulated CCN concentrations which used for evaluation. The MFB and MFE are 28.17% and
34.16%, which meet the suggested benchmark, but the underpredictions of the model estimates are still observed in most areas. Liu et al., (2020) adopted a satellite-based method to retrieve CCN concentrations from 2013 to 2019, which reported that they can reasonably reproduce the spatial pattern of CCN in East Asia. In this study, monthly mean GC-TOMAS CCN concentrations were compared to satellite-retrieved CCN concentrations at supersaturation of approximately 0.2% from Liu et al., (2020). A total of 8 months of satellite-retrieved CCN concentrations were averaged on MERRA-2 grid (corresponding
667 simulation grids) for comparison (Figure S4). The simulated CCN concentrations presented generally similar monthly variations as the satellite-retrieved concentrations, with MFB of 17.23% and MFE of 37.28%, both meet the criteria suggested by Boylan and Russell (2006). GC-TOMAS outputed CCN concentrations (430 cm$^{-3}$) for 8 months were underestimated compared with satellite-retrieved concentrations (587 cm$^{-3}$), and this underestimation is more apparent in July, August, September, and November. However, for other months (February, April, May, and June) the simulated CCN concentrations
slightly underpredicted observations with mean bias (MB) of -75 cm$^{-3}$. This difference is more likely attributable to differences in model performance in different regions. For example, the underpredictions of CCN in May were mainly distributed in eastern coastal area of China, the Korean Peninsula, and Japan. But in August and September, the underprediction of model estimates discrepancies were mainly in the southern and northern part of China, respectively. Due to the limited monitoring data of CCN in our domain during simulation period, we compared predicted results with satellite-retrieved CCN. However,
as Liu et al., (2020) indicated that errors in retrieved data and the CCN counters might cause inaccuracy of satellite CCN inversion results, it was noted that the satellite derived CCN that cannot be treated as true as in-situ observations during validating our model results.

For AOD, monthly averages from the Aerosol Robotic Network (AERONET) Version 3 spectral deconvolution algorithm (SDA) level 2.0 measurements (Giles et al., 2019) were used to validate the model estimations. In total, there are 79
measurements within the simulation domain. Figure S5 displays annual-mean model estimates and AERONET measurements AOD at 550nm (the AERONET AODs at 500nm are converted to 550nm using Ångström exponents at 500nm). The model estimates compared well with measurements with a Pearson's R of 0.84 and only a slightly underprediction of AOD with MBs of -0.12. The respective MFB and MFE were -28.64% and 13.45%, which all meet the benchmark suggested in Boylan and Russel et al, 2006.




### 3.3 Seasonal variations of DMS impacts to $SO_4^{2-}$, CCN, and radiative forcing

By adding of updated DMS emissions (XG-ND), we predicted the enhancement of near-surface $SO_4^{2-}$ concentrations of 0.1-0.3 µg m$^{-3}$ over most areas of seawater (Figure 6(a)). The highest impacts (approximately 0.3 µg m$^{-3}$) occurred in MAM around the South China Sea area due to highest regional mean DMS concentrations in MAM. However, the spatial distributions of
$SO_4^{2-}$ concentrations enhanced by addition of DMS emissions in the four seasons did not exactly follow the spatial and temporal pattern of seawater DMS concentrations (Figure 2). Sea surface wind speed has noticeable impacts on the sea-air DMS flux and followed atmospheric DMS concentrations, which caused higher atmospheric DMS concentrations over the India Ocean in the MAM. Ambient oxidant level also plays an important role in the subsequent DMS oxidation phase. For example, higher atmospheric DMS (300-400pptv) and $SO_2$ (0.2-0.3µg m$^{-3}$) concentrations contributed by DMS can be found around the areas
of East China Sea (Figure S6 and S7) in MAM and JJA. However, a higher contribution of DMS emissions to near-surface $SO_4^{2-}$ concentrations occurred over south China Sea in DJF and MAM. The spatial disparities might be due to the roles of oxidants in the conversion of $SO_2$ into $SO_4^{2-}$ in different seasons, and cloud cover could also affect the aqueous conversion.

The magnitude of the all-sky sulfate DRF at TOA contributed by DMS ranged from -200 to -20 mW m$^{-2}$ in four seasons (Figure 6(b)). The spatial patterns of DRF are highly consistent with those of $SO_4^{2-}$ concentrations, with the stronger negative
DRF (-200 to -120 mW m$^{-2}$) in the areas with higher $SO_4^{2-}$ concentrations contributed by DMS, such as the South China Sea, Philippine Sea, and Japan Sea. It should be noted that DRF calculation is from the whole column of the atmosphere whereas Figure 6(a) just shown the surface layer concentrations, yet the spatial results are still qualitatively similar. As reported by some previous studies (Khan et al., 2016;Chen et al., 2018;Zhao et al., 2021), DMS mainly exists in the lower atmosphere, and impacts of DMS to the $SO_2$ and $SO_4^{2-}$ concentrations are limited to the lower troposphere. So, the magnitude of sulfate
DRF at TOA shown in Figure 6(b) is mostly caused by lower-altitude $SO_4^{2-}$ from DMS. $SO_4^{2-}$ aerosols are non-absorbing aerosols primarily scatter incoming radiation, and the increase in reflected solar radiation flux at TOA and almost equally reduce the radiation at the surface (Ramanathan et al., 2001). Thus, for sulfate aerosol, the magnitude of the cooling effect can be estimated from the aerosol radiative forcing at the TOA. The seasonal mean sulfate DRF has a contribution of -22.24, -18.79, -21.58, and -17.43 mW m$^{-2}$ from DMS over the simulation domain in DJF, MAM, JJA, and SON, respectively. The
magnitude of the DMS-induced sulfate DRF in DJF and JJA is higher than other seasons, but the highest impacts of DMS emissions on $SO_4^{2-}$ concentrations occurred in MAM followed by DJF. The all-sky DRF was calculated based on the RRTMG model using aerosol mass concentrations (whole column) and optical parameters along with surface albedo and cloud fractions from MERRA-2 assimilated meteorological data. Hence, aerosol mass concentrations as well as other parameters can impact the magnitude and spatial distributions of the DRF. For clear sky condition, aerosol scatter more of incoming solar radiation
than in all sky condition, which lead to aerosol DRF at TOA and surface increases compared to all sky conditions.

Figure 7(a) shows the changes in seasonal mean CCN surface concentrations at 0.2% supersaturation (CCN (0.2%)) between the XG - ND simulations. Updated DMS emissions lead to an increase in CCN concentrations by 3 - 42 cm$^{-3}$ over most areas of seawater, and 6 - 16 cm$^{-3}$ in some coastal regions. The highest increases occurred in DJF, followed by MAM.





The impacts of DMS on CCN concentrations are shown in Figure 6(a). The modeled DMS-induced cloud-albedo IRF ranged
from -900 to -100 mW m$^{-2}$ in four seasons (Figure 7(b)), which is much higher relative to that of the sulfate DRF attributable
to DMS. The seasonal mean sulfate IRF had a contribution of -43.29, -45.04, -43.60, and -33.03 mW m$^{-2}$ from DMS in our
domain in DJF, MAM, JJA, and SON, respectively. There are some similarities in the spatial distributions of the effects of
DMS on IRF and CCN. However, the strong negative IRF was mainly over remote oceans ( -900 to -600 mW m$^{-2}$), while as
for CCN, the higher contributions were concentrated within coastal waters. One explanation for these differences was that
strong anthropogenic emissions in Asia leaded to an intense competition for water vapor during cloud-droplet activation, which
further decreased the maximum supersaturation achieved in updrafts and limits droplet activation (Kodros et al., 2016). Also,
the clouds are not necessarily at the heights where CCN changes were affected by DMS.

### 3.4 Annual DMS impacts to SO$_4^{2-}$ , CCN, and radiative forcing

**3.4.1 Annual DMS impacts to SO$_4^{2-}$ , CCN, and radiative forcing between XG and ND simulation**

Figure 8 (a) shows the annual-mean percent changes and absolute changes in SO$_4^{2-}$ and CCN between XG and ND
simulation. Oceanic DMS emissions increased the near-surface SO$_4^{2-}$ and CCN concentrations by 0.1-0.3 μg m$^{-3}$ and 3 - 42
cm$^{-3}$ over most areas of seawater across the four seasons. Due to heavy amounts of anthropogenic pollutants from the continent,
the relative contributions of DMS to SO$_4^{2-}$ and CCN were higher in remote oceanic areas, which reached up to 88% and 42%
of all sources. More than 40% and 20% of the SO$_4^{2-}$ and CCN contributed by DMS emissions were also found in the Philippine
Sea and India Ocean, respectively. The impact of DMS emissions can cover the entire coastal regions of simulation domain,
where DMS had a moderate impact of 0.1-0.18 μg m$^{-3}$ for SO$_4^{2-}$ and 10 -22 cm$^{-3}$ for CCN. Yang et al. (2017) indicated that
DMS emissions only have 20-40% of contributions to SO$_4^{2-}$ concentrations over downwind ocean areas of East Asia, which
was much lower than 40-70% contributions estimated in this study. This discrepancy is mainly ascribed to a significant
reduction (62%) in SO$_2$ emissions in China from 2010 to 2017 (Zheng et al., 2018).

The modeled all sky DRF of DMS induced sulfate here range from -100 to -10 mW m$^{-2}$ (Figure 8 (b)). The sulfate DRF
was the strongest ( -100 to -60 mW m$^{-2}$) over the South China Sea, which is consistent with the distributions of SO$_4^{2-}$
concentrations contributed by DMS emissions. The DMS induced cloud-albedo IRF (-700 to -100 mW m$^{-2}$) here was higher
than the all sky DRF estimate. A relatively strong cooling IRF (-700 to -400 mW m$^{-2}$) induced by DMS emissions can be seen
in the vicinity of equatorial belt in India Ocean and northwest Pacific Ocean. The simulated annual mean sulfate DRF and IRF
is -20.01 and -41.26 mW m$^{-2}$ over the simulation domain, respectively. Li et al. (2019) estimated the annual mean all sky DRF
of -100 mW m$^{-2}$ from DMS emissions over the East China Sea. Our estimates (-20.01 mW m$^{-2}$) were lower than their result,
which is likely attributable to discrepancies in the DMS emissions used to drive the model.





### 3.4.2 Annual DMS impacts to SO₄²⁻, CCN, and radiative forcing between XG and LANA simulation


To quantify the impacts of DMS emissions changes on $SO_4^{2-}$, CCN, and radiative forcing, we compared the XG and LANA simulations (Figure S8(a)). Positive impacts for $SO_4^{2-}$ and CCN can be found in the areas of Indonesia and northwest Pacific Ocean (Figure S8(a)), which was generally consistent with the distribution of changes in DMS emissions flux from XG-LANA (Figure 3b). DMS emissions changes (between XG and LANA) accounted for 4-20% and 6-18% of $SO_4^{2-}$ and CCN

concentrations over areas of Indonesia, and 2-10% and 3-6% of those concentrations over the northwest Pacific Ocean, respectively. The largest decreases were seen in the vicinity of the India Ocean, which was -0.06 μg m⁻³ for $SO_4^{2-}$ and -10 cm⁻³ for CCN. Due to the higher background concentrations contributed by anthropogenic sources, the relative percent change was smaller over that area, where DMS emissions changes only accounted for -8 to -4 % for $SO_4^{2-}$ and -6 to -3 % for CCN. Also, changes of DMS fluxes around the equatorial belt in western Pacific Ocean (Figure 3b) did not directly link to negative

changes in $SO_4^{2-}$ and CCN, which was most likely offset by large scale transport of sulfate caused by DMS from East China Sea. The changes in annual mean DRF and IRF from XG-LANA simulation as shown in Figure S8(b). Negative changes of DRF (-20 to -5 mW m⁻²) mainly were concentrated over the northwest Pacific Ocean. The largest increase in DRF (up to 40 mW m⁻²) was found in the areas of the Japan Sea, and most of the positive changes of DRF (5 to 20 mW m⁻²) were mainly distributed in the region of the Indian Ocean and land areas of India. Their spatial patterns were consistent with distributions

of absolute changes of $SO_4^{2-}$ concentrations. The largest changes in IRF were found in areas of the northwest Pacific Ocean and the Sea of Okhotsk, with changes up to -200 and 200 mW m⁻². The negative changes of IRF from the XG-LANA simulation can span most Pacific Ocean over the simulation domain and some continental regions, and positive changes of IRF are more concentrated within the India Ocean and Sea of Okhotsk. Generally, our estimated sea-air DMS fluxed are lower than those from Lana et al. (2011) over the most of the ocean areas, but the DMS-caused changes to $SO_4^{2-}$, CCN, and radiative forcing

were more varied, with the positive changes over the northwest Pacific Ocean for $SO_4^{2-}$ and CCN, and negative changes in the regions of India Ocean, oppositely, positive changes in the regions of India Ocean for DRF and IRF, and negative changes over the northwest Pacific Ocean.

### 3.5 Limitations of this study


We found several limitations in our emission estimates and modeling study. We try to use machine learning estimates of DMS concentrations to fill the regions without observations. However, the primary weakness of the machine learning method is that the training process is not interpretable and not transparent (Reichstein et al., 2019;Wang et al., 2020). The relationship between training parameters should have a minimal physical interpretation, which should be done in future work to give not only accurate but also credible predictions. While the recently measured 1022 seawater DMS observation data over Chinese

Seas were included training period, for some months (January, November, etc.), there were still not enough data to create a monthly mean. Hence, we temporally averaged input parameters to a seasonal period rather than use monthly data, which is a



limitation of this study, but as shown in Section 3.1, the estimated results showed distinct seasonal variations, and the results are comparable with other studies. Due to the limited continuous measurements of atmospheric DMS and MSA concentrations, we only presented the averaged each cruise survey observations for DMS model evaluation and temporal variation of MSA

prediction performance evaluated only from a single observation site. We acknowledged that this is an important limitation of this study, which prevents us from giving comprehensive estimate (in temporal and spatial scale) of the advantage of our updated DMS emissions. More marine and atmospheric observational data are necessary for further model evaluation.

In addition, due to the limited high temporal resolution monitoring data of CCN for the simulation year 2017 in our domain, we verified the model performance of the CCN simulation by comparing the modeled results with the collected mean

annual observed concentrations of CCN in other previous studies and satellite-retrieved CCN concentrations. We acknowledge that the CCN model-measurement comparisons listed in Table S4 are not the exact times where CCN simulated, and satellite-retrieved CCN (given the uncertainty in water uptake and size distributions) are not necessarily accurate enough to represent real atmospheric CCN concentrations in 2017.

Modeled AOD may be biased during cloudy conditions when AERONET measurements may not be available due to

missing data. Hence, there would be an uncertainty in using monthly averaged measurements and model predictions for comparison.

Different chemical mechanisms of various chemical-transport models and the treatment of aerosol optical properties can also make differences in simulation results. Globally, the annual-mean DRF and IRF contributed by DMS reported by other studies (as listed in Table S5) varied from -0.23 to -0.074 W m$^{-2}$ and -6.55 to -0.3 W m$^{-2}$, respectively. Aerosol-cloud

interactions are a major source of uncertainty in the prediction of climate change, impacting radiative forcing estimates, especially the IRF calculation. Differences in aerosol nucleation schemes, activation parameterizations, and emissions between models can contribute to large discrepancies in their simulation results (Carslaw et al., 2013). However, we did not explore impact of different nucleation schemes on radiative forcing. We recommend that this should be done in the future work to minimize uncertainties in modeling study.

**4 Conclusions**

In this study, we utilized the XGBoost machine learning algorithm to estimate sea-water DMS concentrations by training 12 ocean environmental parameters on newly updated DMS measurements. Recently, 1022 seawater DMS measurements over Chinese Sea were included in our training, and we used the machine learning method to fill the gap at times and in locations without observations. The DMS estimates validation results showed that our XGBoost estimates could capture the observed

DMS concentrations with a correlation coefficient of 0.75. Zonal mean DMS concentrations between 10°S and 30°N latitude areas of simulation domain were higher in MAM than in JJA, and most of ocean area was concentrated in 10°S and 30°N latitude band, which leaded to the highest regional mean DMS concentrations observed in MAM. We estimated annual DMS emission fluxes of 1.25 Tg (S), which accounted for 15.4% of anthropogenic sulfur emissions over the entire simulation domain (covers most of Asia) in 2017. Comparative analysis revealed that the sea-air DMS flux estimated in this study (from XG





estimates) was generally lower than those from global-database DMS emissions (Lana et al., 2011). The model estimates of DMS and MSA from XG simulation, were evaluated by comparing with cruise survey experiments and long-term online measurement site data. In general, the improvement in model performance can be observed by comparing with LANA simulation which uses the global-database DMS emissions.

The modeled DMS-induced sulfate DRF and IRF ranged from -200 to -20 mW m$^{-2}$ and -900 to -100 mW m$^{-2}$ across the
four seasons, respectively. The stronger negative DRF (-120 to -200 mW m$^{-2}$) were in the areas where with higher $SO_4^{2-}$ concentrations contributed by DMS, such as the South China Sea, Philippine Sea, and Japan Sea. However, the strong negative IRF was mainly over remote oceans ( -900 to -600 mW m$^{-2}$), which did not match with the spatial distributions of contributions of DMS to CCN concentrations due to the role of clouds in the IRF. Annually, DMS-induced sulfate IRF (-700 to -100 mW m$^{-2}$) here obviously higher than those all sky DRF (-100 to -10 mW m$^{-2}$). By adding our updated DMS emissions to a simulation
with no DMS (XG-ND), we predict the enhancement of near-surface $SO_4^{2-}$ and CCN concentrations by 0.1-0.3 µg m$^{-3}$ and 3 - 42 cm$^{-3}$, respectively, over most oceanic areas in all four seasons. We found higher contributions from DMS emissions to $SO_4^{2-}$ and CCN in MAM and DJF than JJA and SON.

In this work, we quantified the contributions of DMS to atmospheric $SO_4^{2-}$ and CCN aerosol concentrations along with their radiative effect over a modeled Asian domain (covers most of Asia). This work provides better insights into the source
strength of DMS and its impact on climate, addressing knowledge gaps related to factors controlling aerosols in the marine boundary layer and their climate impacts. As discussed in Section 3.6, there are several limitations that need to be improved in the future work. More marine and atmospheric observational data are necessary for further DMS emission estimates and model evaluation to explore the interactions of DMS with aerosols and radiative forcing. In the future work, we also need to explore the impact of different aerosol nucleation schemes on radiative forcing, to more completely quantify the uncertainties
our modeling study.

*Competing interests*. The authors declare that they have no conflict of interest.


*Acknowledgements.* This work was supported by the National Key Research and Development Program of China (Grant 2016YFA060130X), the Major Program of Shanghai Committee of Science and Technology, China (19DZ1205009), and the National Natural Science Foundation of China (42077195).



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





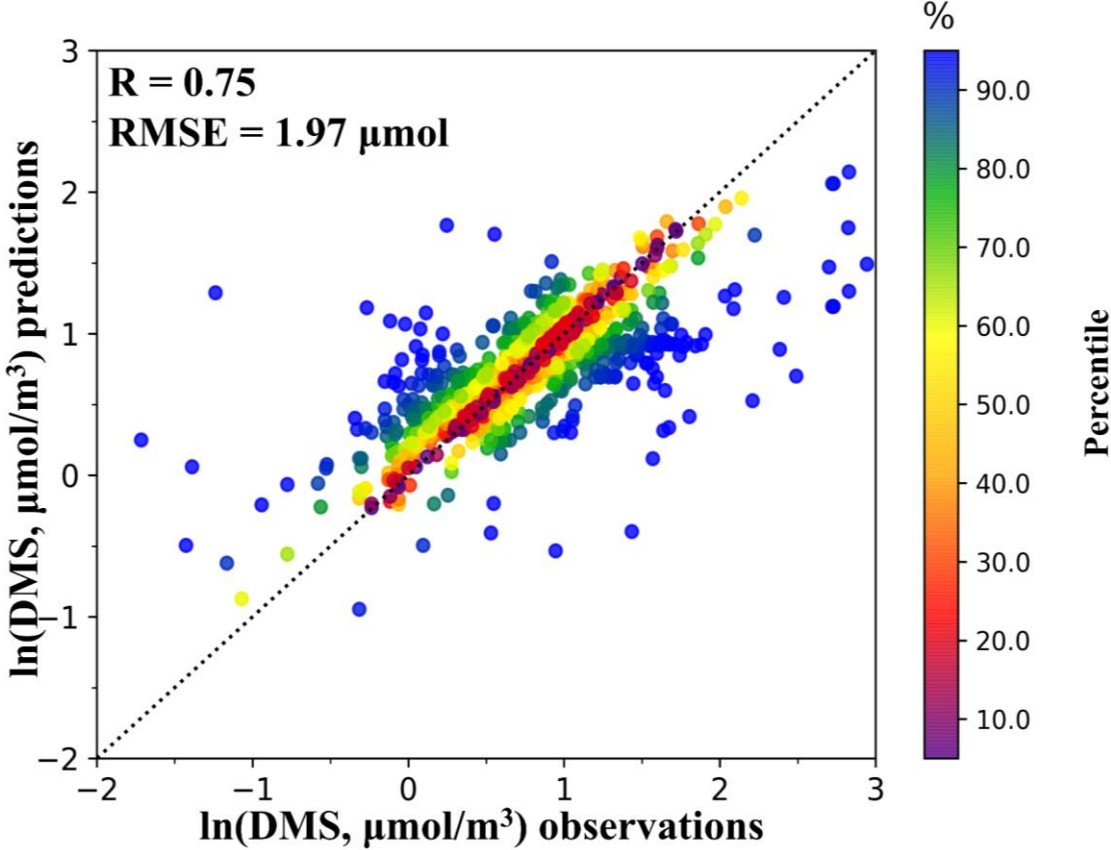

**Figure 1**. The scatter plot compares model predictions and observations of DMS. The colour represents the percentile of distribution of absolute difference between predicted and observation data.

665





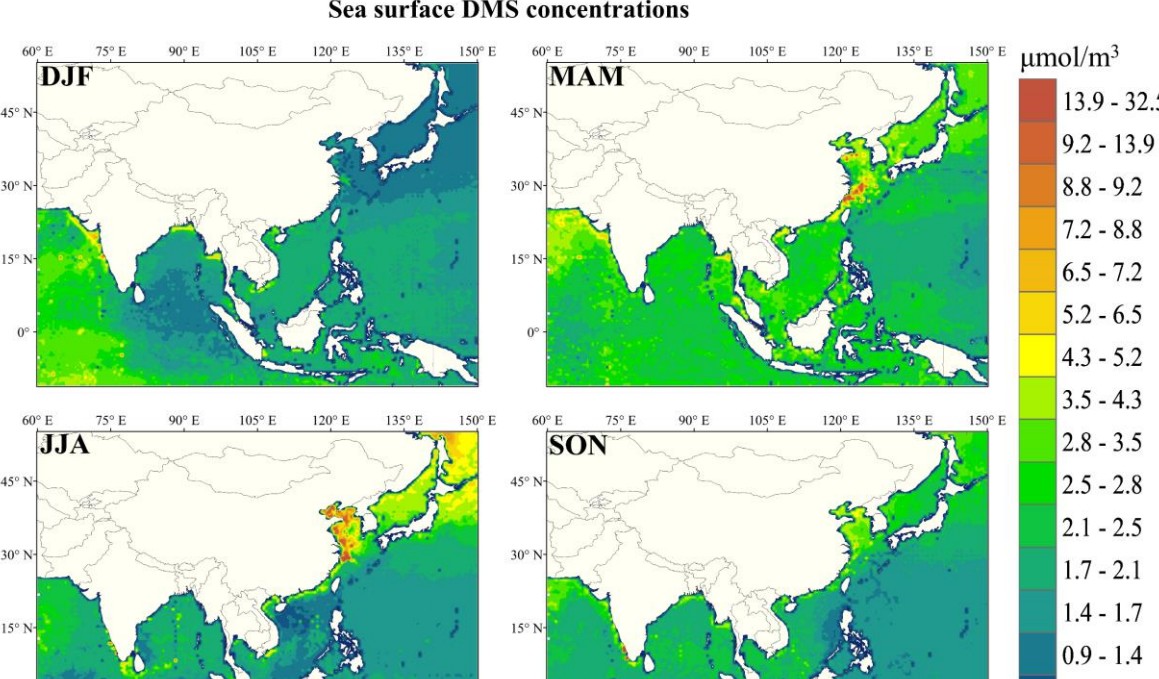

**Figure 2**. Sea-surface DMS concentrations predicted by XGBoost model by season.

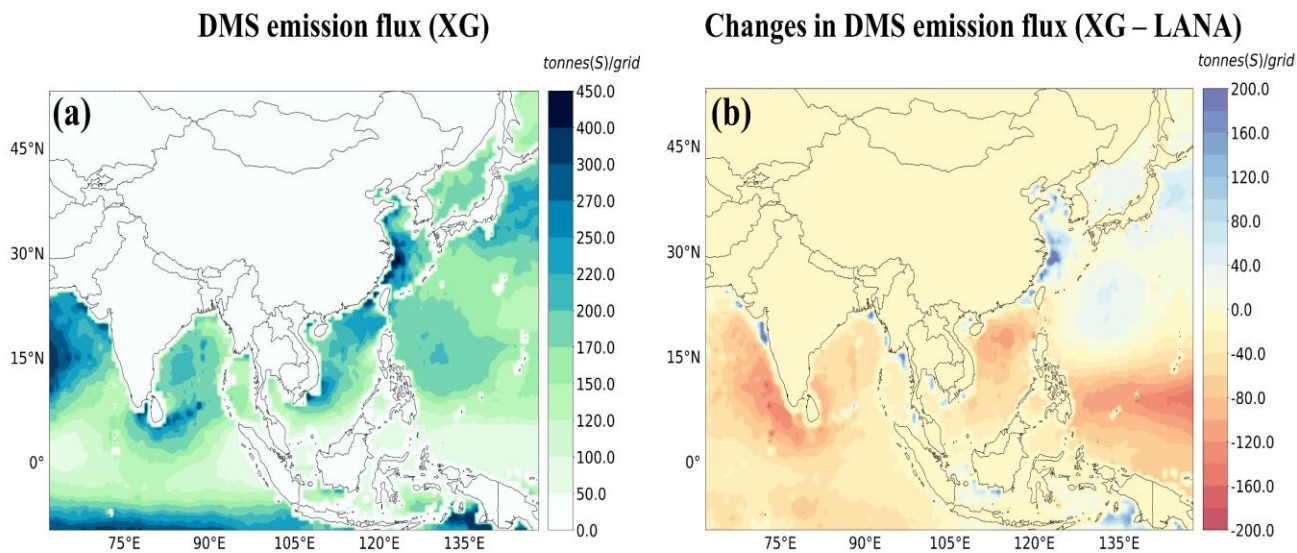

**Figure 3**. Panel (a) presents Annual DMS emission flux calculated based on N00 flux parameterization (Nightingale et al., 2000) from XG sea surface concentrations, panel (b) presents changes between DMS emission flux from updated (XG) and default climatology (LANA).





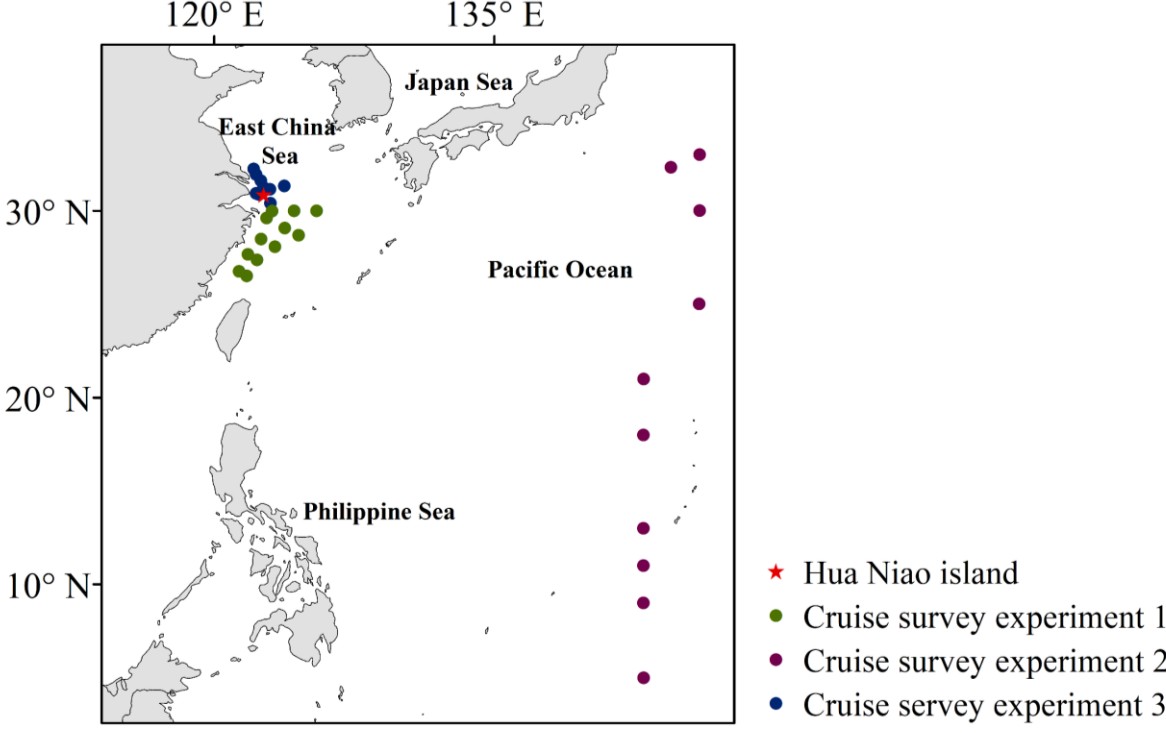

**Figure 4.** Locations of atmospheric DMS observations from cruise survey experiments 1-3, and the MSA observation
675 site in Hua Niao Island.

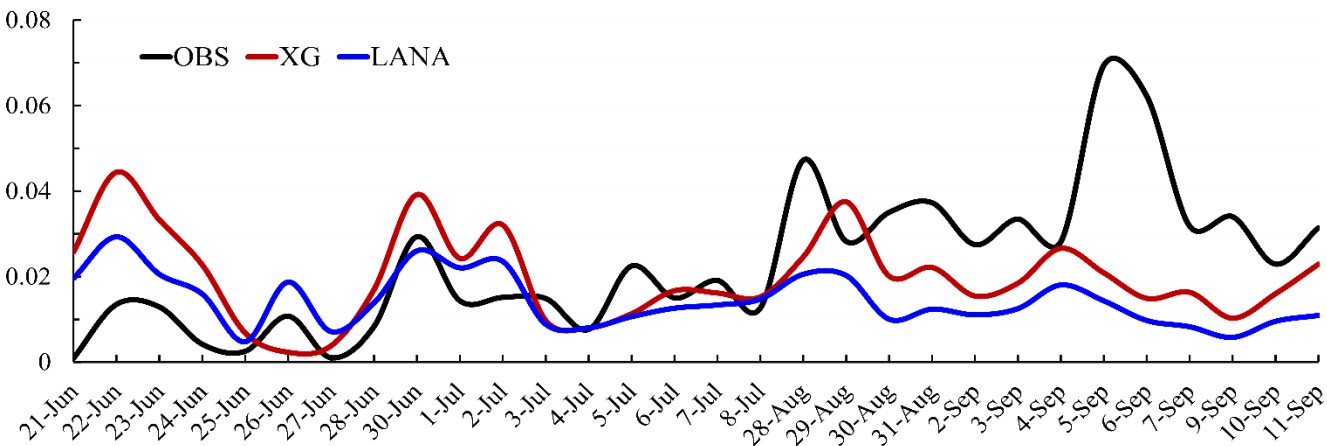

**Figure 5**. A comparison of simulated daily concentrations of MSA with observations at the Hua Niao Island site (Units: µg m⁻³).

680





**Figure 6**. Spatial pattern of the seasonal mean absolute changes in surface SO$_4^{2-}$ (first column) and all-sky DRF (second column) between the XG and ND (no DMS) simulations.





**Absolute contribution of DMS to CCN (XG − ND)**

**DMS-derived IRF (XG − ND)**



**Figure 7**. Spatial pattern of the seasonal mean absolute changes in surface CCN (0.2%) (first column) and cloud-albedo IRF (second column) between the XG and ND (no DMS) simulations.

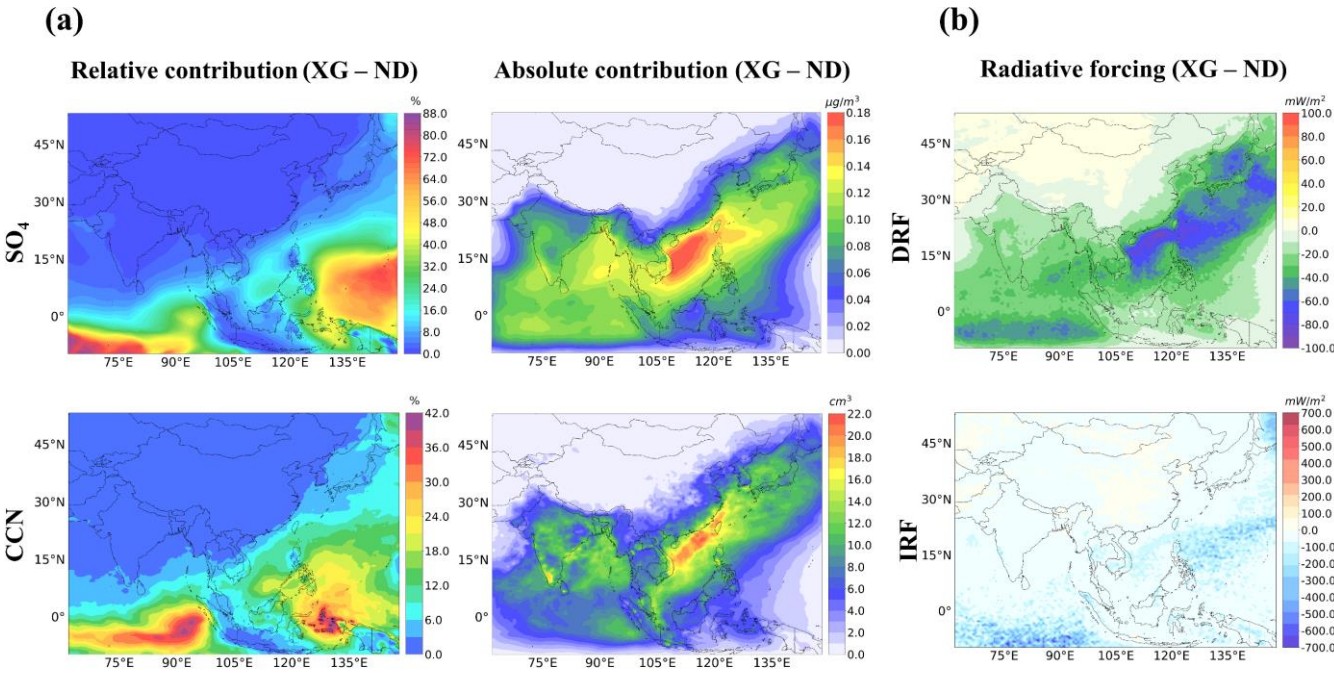

**Figure 8**. Panel (a) presents the spatial distributions of annual mean percent changes and absolute changes in surface $SO_4^{2-}$ and CCN, and panel (b) presents the spatial distributions of annual mean all-sky DRF and cloud-albedo IRF between XG and ND (no DMS) simulations.

**Table 1.** Description of simulation

| Simulation | Description |
| --- | --- |
| XG | DMS emissions on with updated DMS emissions predicted by XGBoost model. |
| LANA | DMS emissions on with default DMS emissions from Lana et al. (2011). |
| ND | DMS emissions turned off. |