# Peer review of "Simulating the radiative forcing of oceanic dimethylsulfide (DMS) in Asia based on machine learning estimates"

_Atmospheric Chemistry and Physics, 2022_

## Author Comment (AC1)

Responses to reviewers' comments

We appreciate the reviewers' careful and thoughtful comments of our manuscript entitled "***Simulating the radiative forcing of oceanic dimethylsulfide (DMS) in Asia based on Machine learning estimates***" and for the many helpful suggestions to improve the article. We have carefully reviewed all comments and revised the article accordingly. The sentences are depicted in yellow in the manuscript text to highlight the new addition and used strikethrough for deletion. To be clear, all the responses are in green background in the below.

**Responses to Reviewer 1 comments:**

1.   My main concern is regarding the training and validation data split. The authors state that "we selected the data from 2° latitude bands between 11°S and 30°N as validation datasets (809 points), while the rest of the data was all used as training data (2939 points)." I am unsure exactly what this means but given the large latitudinal variation in DMS in the region (e.g., Fig. S2) this could introduce unnecessary bias in the model. Why not just use a random split?

**Answer:**

Thank you for your suggestive comments. The reason why we split the validation data manually because it has pointed out that the measurement data collected from the same cruise are highly intercorrelated, and the common practice of shuffling and randomly splitting the data produces an overfitted model due to the validating data can be predicted using near-neighbor values in a recent study which use artificial neural network to generate DMS climatology (Wang et al., 2020). We also have learned from our previous study (Li et al., 2020) that there were a large number of DMS measurements in our simulation domain (Figure S1) were from same shipboard field campaign. So, we followed Wang et al. (2020) selected the validation data manually rather than automatically. As you mentioned, there could be a large latitudinal and seasonal variation in DMS, thus we've also assumed that the in situ sampling times and locations would also influence the effects of model prediction. So, the coordinate space notations (longitude and latitude) and sampling times (months and hours) were also

included as predictors in machine learning estimates. Now we have added a description about these datasets in revised manuscript in the first paragraph of section 2.3:

*"Avoiding the possible large latitudinal and seasonal variation in DMS, the sampling times and geographic coordinates were also included in machine learning estimates. To solve issues in data discontinuity, these datasets were converted to periodic functions as suggested in previous studies (Gade, 2010;Gregor et al., 2017;Wang et al., 2020)."*

*"As suggested by Wang et al. (2020), the measurement data collected from the same cruise are highly intercorrelated, and using near-neighbor values to predict validation data may cause the model overfit. So, we selected the validation data manually rather than automatically."*

2.  While the authors provide a good overall evaluation of their model, given the large seasonal variation in DMS (and it's contributing factors), I would also like to see a validation of the RMSE in each season, even if only in the supplemental. This would provide confidence that the model is providing robust predictions in different regimes.

**Answer:**

Following your suggestion, we have added Table S2 in the supplemental information which illustrated Model performance of DMS concentration in each season, the corresponding description was added to the revised section 2.3 as that *"Model performance for predicting DMS concentration in each season was illustrated Table S2. Predicted DMS concentrations were slightly underestimated in comparison with validation datasets, with mean bias (MB) of -0.59 to -0.21 µmol m$^{-3}$ and normalized mean bias (NMB) of -19.36 to -6.51% across the four seasons. A lower RMSE of 1.81 µmol m$^{-3}$ was observed in spring. The MB and NMB in spring were smaller than those in other seasons, which indicated that model performed best in spring. Most of available validation datasets were concentrated in the spring (about 67.9%). Thus, the imbalanced data may leaded to less ideal performance in other seasons."*

3.  On this point, the authors state that "the training process is not interpretable and not transparent". I would dispute that. One of the benefits of a tree-based model like XGBoost is

that it is quite efficient to investigate the sensitivity of the output to each of the predictors. For example, the authors could provide Shapley values for the different predictors, perhaps in different seasons. I don't believe this is too much work and would help demonstrate the robustness of the model and support the interpretation of their model.

**Answer:**

We have deleted the statement "*However, the primary weakness of the machine learning method is that the training process is not interpretable and not transparent (Reichstein et al., 2019;Wang et al., 2020). the relationship between training parameters should have a minimal physical interpretation, which should be done in future work to give not only accurate but also credible predictions.*" from the section 3.5.

We calculated Shapley values of each predictor across all prediction cases, which displayed in Figure S3. And, the following statements were added to the revised section 2.3 and section 3.1, respectively as the below:

"*Decision-tree-based machine learning model have a high interpretability. The SHapley Additive exPlanation regression (SHAP) (Lundberg et al., 2020) can provide a deeper understanding of model predictions, which allows for individualized feature attribution for every decision. Stirnberg et al. (2021) quantified the impact of various meteorological derivers on $PM_1$ concentrations by using SHAP analysis, and Silva et al. (2022) used SHAP to explore the errors in the prediction of lightning occurrence in a widely used earth system model. In this study, SHAP was applied to investigate the importance of each predictor on model predicted DMS concentrations.*"

"*Figure S3 summarizes the ranked mean SHAP values of each predictor across all prediction cases. The line ranges represent interquartile range across the distribution. Larger SHAP value magnitudes are interpreted to as more important for the prediction task as they have larger contribution from that variable to that prediction. In our study, the most important environmental parameter to predict DMS concentrations was Chl, followed by MLD, PAR, POC, and salinity. Above all, the SHAP value of Chl is more than double its value of MLD*

*and PAR, and much larger than all others. This is consistent with known importance of Chl in developing predicting models of surface water DMS concentrations, because of its biogenic origin (Simó and Dachs, 2002;Galí et al., 2015;Wang et al., 2020;Deng et al., 2021)."*

4. On line 244 the authors state that "Overall, the simulation results of XG in other periods were closer to the observations than those of LANA simulation results." but this doesn't look true by eye. Could the authors report the respective RMSE of each result against the observations for this time series? It does seem that XG represents the variability somewhat better though.

**Answer:**

As reviewer suggested we revised the statement into "*Although in some periods, LANA simulation results were closer to the observations, and XG simulations underpredicted, e.g. RMSE of 0.013 and 0.006 μg m⁻³ for LANA, 0.021 and 0.010 μg m⁻³ for XG during the period of Jun 21 to Jun 25 and Jun 28 to July 3, respectively. However, in the whole, the simulation results of XG in other periods were closer to the observations than those of LANA simulation results, with RMSE of 0.024 and 0.018μg m⁻³ for LANA and XG, respectively*".

5. L24: 'lack of' -> 'lacking'

**Answer:**

Thanks for the correction. We have changed "lack of" to "lacking".

6. L25: The authors state that the DMS emissions flux 'accounts for 15.4% of anthropogenic sulfur emissions', but DMS isn't an anthropogenic source so this doesn't make sense. Perhaps they mean 'equivalent to 15.4% of …'?

**Answer:**

Thanks for the suggestion. As suggested, we have revised the sentences into "which equivalent to 15.4% of anthropogenic sulfur emissions over the entire simulation domain".

7. L30: 'of all sources' -> 'of all sources, respectively'.

**Answer:**

> Thanks for the correction. We have changed "of all sources" to " of all sources, respectively".

8.  L81-84: The sentence which begins "In this study…" is quite long and doesn't seem to make sense, consider rephrasing.

> **Answer:**
>
> As suggested, we have revised the sentence into that "*In this study, we developed the regional DMS emissions for the year 2017 by training eXtreme Gradient Boosting (XGBoost) machine learning algorithms (Chen and Guestrin, 2016) combined with a newly updated dataset*".

9.  L90: The whole manuscript has inappropriate line breaks in the middle of words from this point on, please check.

> **Answer:**
>
> Thanks for the suggestion. We have checked throughout the manuscript, and fixed all the word break issues.

10. L94: " allows modeler" -> "allows modelers"?

> **Answer:**
>
> Thanks for the correction. We have changed "allows modeler" to "allows modelers".

11. L151: "trained XGBoost" -> "trained an XGBoost".

> **Answer:**
>
> Thanks for the correction. We have changed "trained XGBoost" to "trained an XGBoost"

12. L153: "concentrations where without the observations" -> "concentrations in the place of missing observations"?

> **Answer:**
>
> Thanks for the correction. We have changed "concentrations where without the

observations" to "concentrations in the place of missing observations".

13. L215: "which is 15.4%" -> "which corresponds to 15.4%"?

**Answer:**

Thanks for the correction. We have changed "which is 15.4%" to "which corresponds to 15.4%".

14. L394-396: This is true and the authors might like to cite Schutgens et al. 2016 (https://acp.copernicus.org/articles/16/1065/2016/) in support of the claim.

**Answer:**

We have added the reference to the manuscript and updated citation at the third paragraph of section 3.5.

15. Could the authors use the same colour scale for IRF and DRF in Figures 7 and 8 to aid comparison?

**Answer:**

As reviewer suggested, we have used the colour bar for DRF regenerate the Fugure 7b and 8b, and now is the same colour scale for IRF and DRF in Figures 7 and 8.

**References**

Cao, D., Ma, Y., Sun, L., and Gao, L.: Fast observation simulation method based on XGBoost for visible bands over the ocean surface under clear-sky conditions, Remote Sensing Letters, 12, 674-683, 10.1080/2150704X.2021.1925371, 2021.

Gade, K.: A Non-singular Horizontal Position Representation, Journal of Navigation, 63, 395-417, 10.1017/S0373463309990415, 2010.

Gregor, L., Kok, S., and Monteiro, P. M. S.: Empirical methods for the estimation of Southern Ocean CO2: support vector and random forest regression, Biogeosciences, 14, 5551-5569, 10.5194/bg-14-5551-2017, 2017.

Ivatt, P. D., and Evans, M. J.: Improving the prediction of an atmospheric chemistry transport model using gradient-boosted regression trees, Atmos. Chem. Phys., 20, 8063-8082, 10.5194/acp-20-8063-2020, 2020.

Li, S., Sarwar, G., Zhao, J., Zhang, Y., Zhou, S., Chen, Y., Yang, G., and Saiz-Lopez, A.: Modeling the Impact of Marine DMS Emissions on Summertime Air Quality Over the Coastal East China Seas, Earth and Space Science, 7, e2020EA001220, https://doi.org/10.1029/2020EA001220, 2020.

Lundberg, S. M., Erion, G., Chen, H., DeGrave, A., Prutkin, J. M., Nair, B., Katz, R., Himmelfarb, J., Bansal, N., and Lee, S.-I.: From local explanations to global understanding with explainable AI for trees, Nature Machine Intelligence, 2, 56-67, 10.1038/s42256-019-0138-9, 2020.

Pan, B.: Application of XGBoost algorithm in hourly PM2.5 concentration prediction, IOP Conference Series: Earth and Environmental Science, 113, 012127, 10.1088/1755-1315/113/1/012127, 2018.

Qian, Q. F., Jia, X. J., and Lin, H.: Machine Learning Models for the Seasonal Forecast of Winter Surface Air Temperature in North America, Earth and Space Science, 7, e2020EA001140, https://doi.org/10.1029/2020EA001140, 2020.

Shwartz-Ziv, R., and Armon, A.: Tabular data: Deep learning is not all you need, Information Fusion, 81, 84-90, https://doi.org/10.1016/j.inffus.2021.11.011, 2022.

Silva, S. J., Keller, C. A., and Hardin, J.: Using an Explainable Machine Learning Approach to Characterize Earth System Model Errors: Application of SHAP Analysis to Modeling Lightning Flash Occurrence, Journal of Advances in Modeling Earth Systems, 14, e2021MS002881, https://doi.org/10.1029/2021MS002881, 2022.

Stirnberg, R., Cermak, J., Kotthaus, S., Haeffelin, M., Andersen, H., Fuchs, J., Kim, M., Petit, J. E., and Favez, O.: Meteorology-driven variability of air pollution (PM1) revealed with explainable machine learning, Atmos. Chem. Phys., 21, 3919-3948, 10.5194/acp-21-3919-2021, 2021.

Sun, Y., Yin, H., Lu, X., Notholt, J., Palm, M., Liu, C., Tian, Y., and Zheng, B.: The drivers and health risks of unexpected surface ozone enhancements over the Sichuan Basin, China, in 2020,

Atmos. Chem. Phys., 21, 18589-18608, 10.5194/acp-21-18589-2021, 2021.

Wang, W. L., Song, G., Primeau, F., Saltzman, E. S., Bell, T. G., and Moore, J. K.: Global ocean dimethyl sulfide climatology estimated from observations and an artificial neural network, Biogeosciences, 17, 5335-5354, 10.5194/bg-17-5335-2020, 2020.

Zamani Joharestani, M., Cao, C., Ni, X., Bashir, B., and Talebiesfandarani, S.: PM2.5 Prediction Based on Random Forest, XGBoost, and Deep Learning Using Multisource Remote Sensing Data, Atmosphere, 10, 10.3390/atmos10070373, 2019.

Zhang, R., Li, B., and Jiao, B.: Application of XGboost Algorithm in Bearing Fault Diagnosis, IOP Conference Series: Materials Science and Engineering, 490, 072062, 10.1088/1757-899x/490/7/072062, 2019.

---

## Author Comment (AC2)

**Responses to reviewers' comments**

We appreciate the reviewers' careful and thoughtful comments of our manuscript entitled "***Simulating the radiative forcing of oceanic dimethylsulfide (DMS) in Asia based on Machine learning estimates***" and for the many helpful suggestions to improve the article. We have carefully reviewed all comments and revised the article accordingly. The sentences are depicted in yellow in the manuscript text to highlight the new addition and used strikethrough for deletion. To be clear, all the responses are in green background in the below.

**Responses to Reviewer 2 comments:**

1. The authors need to explain in more detail why they chose the XGBoost model instead of a different ML model and should give further details on the performance of this approach, both at training and at validation, rather than only Pearson's coefficient and RMSE, especially when the RMSE is of the same order of magnitude as the predicted concentrations. I would have liked to see other performance metrics as well, such as relative errors.

> **Answer:**
>
> Thanks for your suggestions. As reviewer suggested, we have added Table S2 in the supplemental information which illustrated other performance metrics for DMS predictions in each season, and also added to the second paragraph of section 2.3 that *"Model performance for predicting DMS concentration in each season was illustrated Table S2. Predicted DMS concentrations were slightly underestimated in comparison with validation datasets, with mean bias (MB) of -0.59 to -0.21 μmol m$^{-3}$ and normalized mean bias (NMB) of -19.36 to -6.51% across the four seasons. A lower RMSE of 1.81 μmol m$^{-3}$ was observed in spring. The MB and NMB in spring were smaller than those in other seasons, which indicated that model performed best in spring. Most of available validation datasets were concentrated in the spring (about 67.9%). Thus, the imbalanced data may leaded to less ideal performance in other seasons."*

The reason why we chose the XGBoost model, due to its advantages of scalability, computing efficiency and prediction accuracy, and robust to randomness, it has also been widely used in geoscience, predictions of atmospheric composition, and other areas (Sun et al., 2021;Ivatt and Evans, 2020;Qian et al., 2020;Zhang et al., 2019;Silva et al., 2022;Cao et al., 2021). Some studies also showed that XGBoost consistently outperforms other ML algorithms (Zamani Joharestani et al., 2019;Pan, 2018). Among the ML approaches, some deep learning techniques tend to require larger amounts of training data to make reasonable predictions, whereas Xgboost is good for tabular data with a small number of variables (Qian et al., 2020;Shwartz-Ziv and Armon, 2022). In our study, the total number of training samples is 2939, thus, we believe that ML model like XGBoost requiring small training data sets is preferred in our DMS concentration predicting experiments. So, we selected XGBoost model to estimate DMS concentrations.

We also added the introduction of advantages of XGBoost in updated section 2.3 that "*XGBoost (machine learning algorithm under the Gradient Boosting framework) was used due to its many advantages. For example, XGBoost is computationally efficiency, has prediction accuracy, requires less tunning, and is scalable, has been widely used in area of geoscience (Sun et al., 2021;Ivatt and Evans, 2020;Pan, 2018;Qian et al., 2020;Silva et al., 2022;Cao et al., 2021), and generally outperformed other models. Moreover, Xgboost is good for tabular data and does not require large training datasets (Shwartz-Ziv and Armon, 2022). Thus, to better capture the nonlinear relationship between DMS and the parameters that influence it, we trained an XGBoost model with the entire dataset to predict sea surface DMS concentrations where without the observations in the place of missing observations*".

2. I am also puzzled by the high correlation coefficient and small RMSE in Figure S5, where observations are compared against model predictions for AOD. It is clear that the points are not around the 1:1 line (it looks like that the slope of the fitted straight line is of the order of 0.6). How can R be 0.84 then?? Can the calculations be rechecked please?

**Answer:**

We have rechecked data extraction and calculation process and regenerated the Figure S5(Figure S6 now), and the results showed that the correlation coefficient R is indeed equal to 0.84 ($R^2$=0.7094), and please see the below table for the original data with 78 data points, which there were 79 data points included in Figure S5 (now is Figure S6) before revision. The equation for the fitted line is updated from y=0.6x+0.0057 to y=0.57x+0.0151, the MB and RMSE are changed from 0.123 and 0.164 to 0.128 and 0.169. All the decriptions have been revised accordingly in section 3.2.2.

**Table 1**. Comparison of annual mean modelled AOD concentrations with observations.

| aod_sim | aod_obs |
|---|---|
| 0.198019 | 0.413011 |
| 0.055318 | 0.113226 |
| 0.071594 | 0.160666 |
| 0.111674 | 0.326399 |
| 0.025943 | 0.08624 |
| 0.046879 | 0.117806 |
| 0.070258 | 0.135001 |
| 0.161068 | 0.286322 |
| 0.070329 | 0.137849 |
| 0.120607 | 0.218007 |
| 0.035525 | 0.162991 |
| 0.18705 | 0.372478 |
| 0.11299 | 0.184612 |
| 0.134788 | 0.224337 |
| 0.119421 | 0.119859 |
| 0.358123 | 0.49824 |
| 0.103856 | 0.168056 |
| 0.244979 | 0.33077 |
| 0.115247 | 0.203173 |
| 0.154072 | 0.265692 |
| 0.14188 | 0.240256 |
| 0.197643 | 0.204565 |
| 0.313507 | 0.575809 |

| | |
|---|---|
| 0.155677 | 0.14272 |
| 0.198349 | 0.290638 |
| 0.19668 | 0.30869 |
| 0.261558 | 0.301686 |
| 0.283822 | 0.280422 |
| 0.140958 | 0.536407 |
| 0.3855 | 0.736027 |
| 0.302966 | 0.437711 |
| 0.228412 | 0.417132 |
| 0.246119 | 0.578429 |
| 0.143475 | 0.05651 |
| 0.524572 | 0.742941 |
| 0.226769 | 0.596238 |
| 0.146769 | 0.461829 |
| 0.178 | 0.398713 |
| 0.154861 | 0.357078 |
| 0.267373 | 0.347199 |
| 0.615157 | 0.720752 |
| 0.325735 | 0.469155 |
| 0.535509 | 0.728547 |
| 0.607197 | 0.670223 |
| 0.351102 | 0.658105 |
| 0.340464 | 0.416898 |
| 0.192791 | 0.029187 |
| 0.023108 | 0.045445 |
| 0.015623 | 0.035831 |
| 0.321394 | 0.529831 |
| 0.475251 | 0.736571 |
| 0.240055 | 0.259574 |
| 0.176538 | 0.281486 |
| 0.123795 | 0.144617 |
| 0.462741 | 0.686936 |
| 0.13623 | 0.228129 |
| 0.230922 | 0.501796 |
| 0.174646 | 0.181009 |
| 0.177354 | 0.240868 |
| 0.178796 | 0.222414 |
| 0.12677 | 0.208805 |
| 0.249448 | 0.334583 |
| 0.271836 | 0.327218 |
| 0.250165 | 0.428985 |
| 0.166509 | 0.148778 |

| | |
|---|---|
| 0.249105 | 0.435451 |
| 0.171069 | 0.28323 |
| 0.156291 | 0.21485 |
| 0.051773 | 0.307126 |
| 0.271465 | 0.591433 |
| 0.274265 | 0.617702 |
| 0.232098 | 0.599271 |
| 0.096996 | 0.172965 |
| 0.047065 | 0.138489 |
| 0.127681 | 0.191397 |
| 0.022924 | 0.101644 |
| 0.147541 | 0.253835 |
| 0.009818 | 0.14474 |

3.  As a final comment, the language in the paper needs to be checked, since the paper has a few grammatical errors.

**Answer:**

Thanks for the suggestion. We have checked and revised all the grammar and wording issues throughout the manuscript.

We appreciate for Editors/Reviewers' careful and thoughtful appraisal of our work and for the many helpful suggestions. We hope that the corrections in response to your feedback will meet with approval.

**References**

Cao, D., Ma, Y., Sun, L., and Gao, L.: Fast observation simulation method based on XGBoost for visible bands over the ocean surface under clear-sky conditions, Remote Sensing Letters, 12, 674-683, 10.1080/2150704X.2021.1925371, 2021.

Gade, K.: A Non-singular Horizontal Position Representation, Journal of Navigation, 63, 395-417, 10.1017/S0373463309990415, 2010.

Gregor, L., Kok, S., and Monteiro, P. M. S.: Empirical methods for the estimation of Southern Ocean $CO_2$: support vector and random forest regression, Biogeosciences, 14, 5551-5569, 10.5194/bg-14-5551-2017, 2017.

Ivatt, P. D., and Evans, M. J.: Improving the prediction of an atmospheric chemistry transport model using gradient-boosted regression trees, Atmos. Chem. Phys., 20, 8063-8082, 10.5194/acp-20-8063-2020, 2020.

Li, S., Sarwar, G., Zhao, J., Zhang, Y., Zhou, S., Chen, Y., Yang, G., and Saiz-Lopez, A.: Modeling the Impact of Marine DMS Emissions on Summertime Air Quality Over the Coastal East China Seas, Earth and Space Science, 7, e2020EA001220, https://doi.org/10.1029/2020EA001220, 2020.

Lundberg, S. M., Erion, G., Chen, H., DeGrave, A., Prutkin, J. M., Nair, B., Katz, R., Himmelfarb, J., Bansal, N., and Lee, S.-I.: From local explanations to global understanding with explainable AI for trees, Nature Machine Intelligence, 2, 56-67, 10.1038/s42256-019-0138-9, 2020.

Pan, B.: Application of XGBoost algorithm in hourly PM2.5 concentration prediction, IOP Conference Series: Earth and Environmental Science, 113, 012127, 10.1088/1755-1315/113/1/012127, 2018.

Qian, Q. F., Jia, X. J., and Lin, H.: Machine Learning Models for the Seasonal Forecast of Winter Surface Air Temperature in North America, Earth and Space Science, 7, e2020EA001140, https://doi.org/10.1029/2020EA001140, 2020.

Shwartz-Ziv, R., and Armon, A.: Tabular data: Deep learning is not all you need, Information Fusion, 81, 84-90, https://doi.org/10.1016/j.inffus.2021.11.011, 2022.

Silva, S. J., Keller, C. A., and Hardin, J.: Using an Explainable Machine Learning Approach to Characterize Earth System Model Errors: Application of SHAP Analysis to Modeling Lightning Flash Occurrence, Journal of Advances in Modeling Earth Systems, 14, e2021MS002881, https://doi.org/10.1029/2021MS002881, 2022.

Stirnberg, R., Cermak, J., Kotthaus, S., Haeffelin, M., Andersen, H., Fuchs, J., Kim, M., Petit, J. E., and Favez, O.: Meteorology-driven variability of air pollution (PM1) revealed with explainable machine learning, Atmos. Chem. Phys., 21, 3919-3948, 10.5194/acp-21-3919-2021, 2021.

Sun, Y., Yin, H., Lu, X., Notholt, J., Palm, M., Liu, C., Tian, Y., and Zheng, B.: The drivers and health risks of unexpected surface ozone enhancements over the Sichuan Basin, China, in 2020,

Atmos. Chem. Phys., 21, 18589-18608, 10.5194/acp-21-18589-2021, 2021.

Wang, W. L., Song, G., Primeau, F., Saltzman, E. S., Bell, T. G., and Moore, J. K.: Global ocean dimethyl sulfide climatology estimated from observations and an artificial neural network, Biogeosciences, 17, 5335-5354, 10.5194/bg-17-5335-2020, 2020.

Zamani Joharestani, M., Cao, C., Ni, X., Bashir, B., and Talebiesfandarani, S.: PM2.5 Prediction Based on Random Forest, XGBoost, and Deep Learning Using Multisource Remote Sensing Data, Atmosphere, 10, 10.3390/atmos10070373, 2019.

Zhang, R., Li, B., and Jiao, B.: Application of XGboost Algorithm in Bearing Fault Diagnosis, IOP Conference Series: Materials Science and Engineering, 490, 072062, 10.1088/1757-899x/490/7/072062, 2019.

---

## Author Response (AR2)

**Responses to Editor's comments**

We appreciate the Editor's comments of our manuscript entitled "***Simulating the radiative forcing of oceanic dimethylsulfide (DMS) in Asia based on Machine learning estimates***". To be clear, all the responses are in green background in the below.

**Comments to the author:**

1. Thank you very much for the revision of your manuscript which can be now accepted for publication in ACP after a few small corrections (lines refer to the track changes pdf):

   1) line 26: is equivalent to

   2) line 87: replace ' combined with' by ' using '

   3) line 159: efficient ... tuning, is scalable,...

   4) line 196: in Table S2

   5) line 200: in spring (remove the)

   Please profit from this request to read once again your manuscript for potential small corrections needed.

   > **Answer:**
   >
   > Thanks for the correction. All the requests have been revised accordingly, and we have checked all the grammar and wording issues throughout the manuscript.